# Speckle-modulating optical coherence tomography in living mice and humans

Orly Liba[1,2,3,4], Matthew D. Lew[1], Elliott D. SoRelle[1,3,4,5], Rebecca Dutta[1,3,4], Debasish Sen[1,3,4], Darius M. Moshfeghi[6], Steven Chu[4,5,7] & Adam de la Zerda[1,2,3,4,5,8]

Optical coherence tomography (OCT) is a powerful biomedical imaging technology that relies on the coherent detection of backscattered light to image tissue morphology *in vivo*. As a consequence, OCT is susceptible to coherent noise (speckle noise), which imposes significant limitations on its diagnostic capabilities. Here we show speckle-modulating OCT (SM-OCT), a method based purely on light manipulation that virtually eliminates speckle noise originating from a sample. SM-OCT accomplishes this by creating and averaging an unlimited number of scans with uncorrelated speckle patterns without compromising spatial resolution. Using SM-OCT, we reveal small structures in the tissues of living animals, such as the inner stromal structure of a live mouse cornea, the fine structures inside the mouse pinna, and sweat ducts and Meissner's corpuscle in the human fingertip skin—features that are otherwise obscured by speckle noise when using conventional OCT or OCT with current state of the art speckle reduction methods.

[1] Department of Structural Biology, Stanford University, Stanford, California 94305, USA. [2] Department of Electrical Engineering, Stanford University, Stanford, California 94305, USA. [3] Molecular Imaging Program at Stanford, Stanford, California 94305, USA. [4] The Bio-X Program, Stanford, California 94305, USA. [5] Biophysics Program at Stanford, Stanford, California 94305, USA. [6] Department of Ophthalmology, Byers Eye Institute, Stanford University School of Medicine, Palo Alto, California 94303, USA. [7] Departments of Physics and Molecular and Cellular Physiology, Stanford University, Stanford, California 94305, USA. [8] The Chan Zuckerberg Biohub, San Francisco, California 94158, USA. Correspondence and requests for materials should be addressed to A.d.l.Z. (email: adlz@stanford.edu).

Since its initial demonstration nearly 25 years ago[1], optical coherence tomography (OCT) has become widely used by ophthalmologists for diagnosis of eye diseases[2]. Recently, OCT has gained popularity for its diagnostic capabilities in cardiology[3], dermatology[4–6], dentistry[7] and cancer research[8–10]. Because of the nature of imaging with coherent light, OCT suffers from speckle noise[11] that effectively causes significant degradation in spatial resolution and prevents the imaging technique from achieving greater diagnostic potential. Speckle noise is inherent to all coherent imaging methodologies and arises from the interference of light scattered from multiple points within a turbid sample[12], such as biological tissue. Following the initial development of OCT, researchers have described various techniques for reducing speckle noise. One group of methods involves incoherently averaging (compounding) several images with uncorrelated speckle noise. Obtaining images with non-correlated speckle patterns can be achieved with various acquisition schemes, including the following: scanning from different angles, scanning several adjacent images, scanning with bands of different incident wavelengths and scanning with different polarizations. These methods are referred to as angular[13], spatial[14], frequency[15] and polarization compounding[11]. The two basic limitations of these current methods are, first, that increasing compromises in resolution are required to further decrease speckle noise, and, second, that the number of uncorrelated speckle patterns is constrained. Hence, these approaches can never eliminate speckle noise entirely. The second group of methods to reduce speckle noise is based on image-processing techniques such as adaptive filters[16] and wavelet analysis[17], among others[18–20]. These methods cannot reveal information that was lost because of speckle; rather, they merely reduce the appearance of speckle noise. Achieving speckle reduction through the use of a partially spatially coherent source[21] has been suggested in the past for OCT imaging; to date, such a source has not been demonstrated for speckle reduction in tomograms of turbid media.

Speckle noise also poses significant challenges outside the field of OCT, and different methods to reduce speckle have been attempted for distinct applications. For example, the use of partially coherent illumination, implemented by a moving diffuser in the optical path, has been previously explored for imaging[12,22,23], display[24,25] and holography[26,27]. In some cases speckle can be utilized to improve imaging, as in quantitative phase microscopy[28], holographic microscopy[29] and wide-field microscopy[30]. In OCT, the variation of speckle is extremely useful for detecting and measuring flow[31,32].

In contrast to prior speckle reduction methods for OCT, the technique presented here, speckle-modulating OCT (SM-OCT), can be used to acquire an unlimited number of uncorrelated speckle patterns and effectively remove speckle noise without degrading the resolution of the image. Hence, SM-OCT clarifies and reveals structures that are otherwise obscured or undetectable. The following section describes our implementation of SM-OCT along with a theoretical model of speckle reduction. Next, we demonstrate the ability of SM-OCT to increase the effective image resolution and visibility in two types of phantoms versus traditional OCT. We present a statistical analysis of signal values from OCT and SM-OCT images and show how increasing the number of scan averages changes the signal statistics from speckle statistics to a distribution that better describes the statistics of the phantom. We continue by confirming the expected functional dependence of the speckle contrast on the number of averaged scans. Finally, we provide demonstrations of SM-OCT for imaging the tissues of living subjects: a mouse retina and cornea, a mouse ear pinna and human fingertip skin. In all examples, SM-OCT was able to reveal fine structures not previously observed with such clarity when conventional OCT was used.

## Results

**Theory**. The fundamental concept of SM-OCT is the introduction of time-varying local phase shifts within the light beam illuminating the sample. These variations translate into local phase shifts in the light reflected from scatterers within each resolution element (voxel), which subsequently yield non-correlated speckle patterns that can be incoherently averaged over time to create an image with reduced speckle noise (Fig. 1a). Because each image is acquired at the same angle, sample position and set of illumination wavelengths, increasing the number of compounded images does not lead to an inherent degradation in resolution. Because increasing the number of uncorrelated images does not reduce resolution, it is possible to average many images together and subsequently reduce speckle noise such that it is undetectable relative to other noise sources in the image.

An approximate mathematical description of this phenomenon is given by equation (1):

$$I = \frac{1}{M}\sum_{m=1}^{M}\left|\sum_{n=1}^{N} a_n e^{i\varphi_n} e^{i\theta_{n,m}}\right| \tag{1}$$

in which $I$ is the pixel value after averaging $M$ images obtained at different times and with different local phase shifts within the illuminating beam. $N$ is the number of scatterers inside a voxel. For each scatterer $n$ within that voxel, $a_n$ is the scattering amplitude (proportional to its amplitude reflection coefficient) and $\varphi_n$ is the phase delay due to the axial location of the scatterer. $\theta_{n,m}$ is the local phase shift of the illumination beam, which changes in time in SM-OCT, at the location of scatterer $n$. Simulations show how this approach decreases speckle noise in Supplementary Note 1 and Supplementary Fig. 1.

**Implementation and model**. The implementation of SM-OCT is straightforward and does not require specialized equipment or light sources. We describe here a method to adapt any OCT system as a SM-OCT system. We have demonstrated SM-OCT on two commercial spectral domain OCT (SDOCT) systems: a high-resolution (HR-OCT) skin-imager (Ganymede HR, Thorlabs) and a clinical retinal imager (iFusion, Optovue). For both devices, we implemented local, random time-varying phase shifts by translating or rotating a ground glass diffuser (Thorlabs) in an OCT conjugate image plane (Fig. 1b and Supplementary Fig. 2). In both systems, the diffuser is moved in a plane perpendicular to the optical axis by either a translation or rotation motor (Z812, Thorlabs and RSC-103, Pacific Laser Equipment, respectively). The image is acquired several times, imaging the same exact location of the sample but through different locations on the diffuser. The random time-varying thickness pattern of the diffuser changes the speckle pattern of the image such that each frame has a different speckle pattern. After averaging $M$ measurements, speckle noise decreases by a factor of $\sqrt{M}$ (ref. 11); for example, a mere nine averages will lead to a threefold reduction in speckle noise.

The expected performance of a diffuser in SM-OCT along with the necessary displacement of the diffuser can be derived from a model of the sample arm, as shown in Supplementary Note 2. In this model the diffuser contributes a locally varying phase to the beam propagating through the diffuser to and from the sample. The phase variations result from the difference in the refractive index of glass and air, $\Delta n$, and the varying thickness of the ground glass, $d(x_d + x, y_d + y)$, with $(x_d, y_d)$ representing the location of the centre of the beam on the diffuser and $(x, y)$ representing the position relative to the centre of the OCT beam.

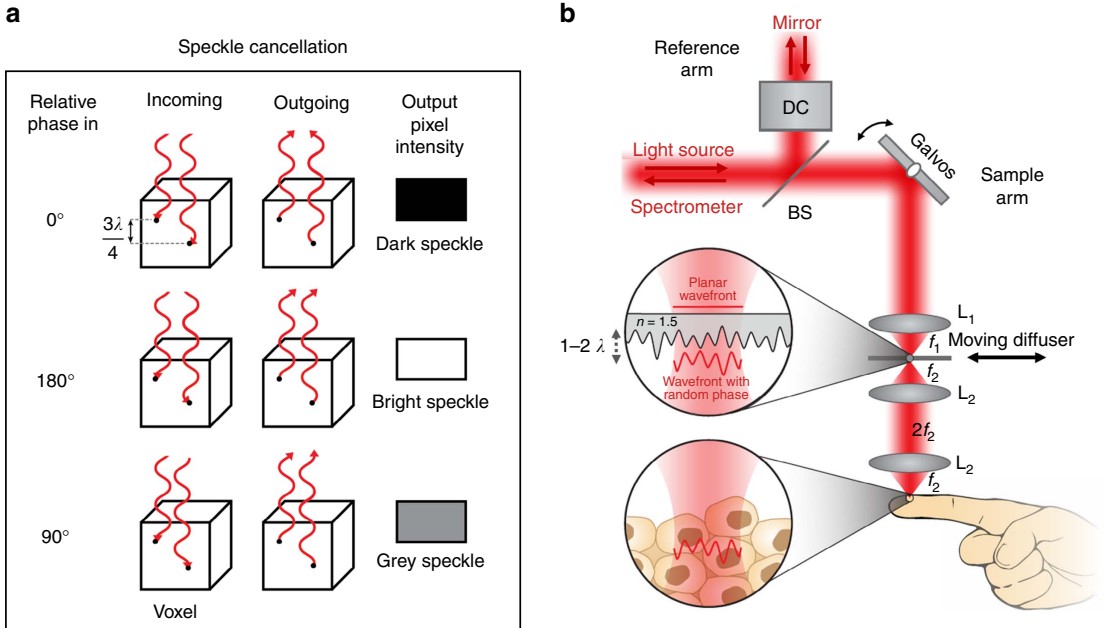

**Figure 1 | SM-OCT speckle removal concept and implementation.** (**a**) Introducing local phase shifts between scatterers in the same voxel changes the intensity of the resulting speckle noise, enabling one to reduce speckle noise via averaging many different phase shifts. This leads to the detection of scatterers otherwise hidden by the speckle noise. (**b**) Implementation of SM-OCT on the high-resolution OCT system. DC, dispersion compensation; BS, beam splitter; $L_1$, lens of the conventional OCT; $L_2$, lenses added to create a $4f$ imaging system; $f_1$, focal length of $L_1$; $f_2$, focal length of $L_2$; $n$, refractive index of the diffuser; $\lambda$, the centre wavelength of the light source.

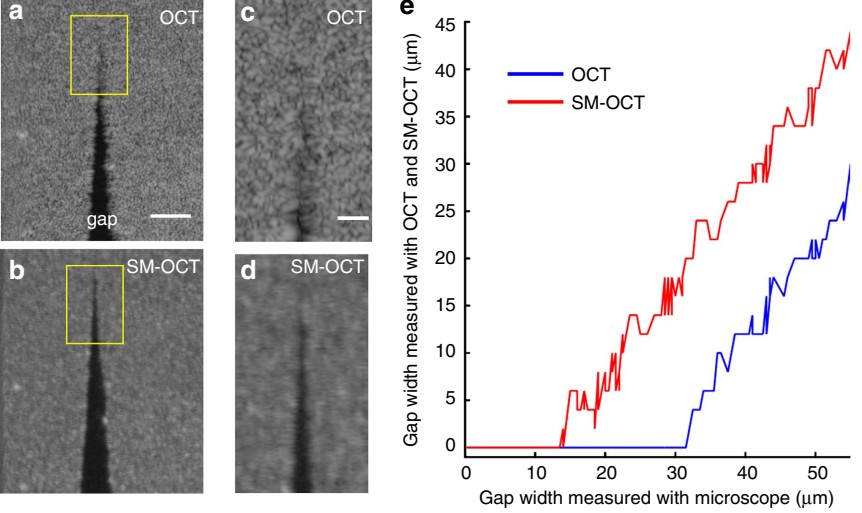

**Figure 2 | SM-OCT demonstration of improved visibility of closely spaced scattering objects.** (**a,b**) A phantom composed of PDMS and $TiO_2$ powder was shaped to form a gap of decreasing size to evaluate the effective spatial resolution of SM-OCT versus OCT. The images shown here are *en face* OCT (**a**) and SM-OCT (**b**) scans inside the phantom. Scale bar, 100 μm. (**c,d**) Close-up view on the regions marked in **a,b** showing the micron-size gap. Scale bar, 50 μm. The gap that is clearly visible in SM-OCT does not appear in the OCT image due to speckle noise. (**e**) The size of the gap measured from the OCT and SM-OCT images versus the size of the gap measured from a bright-field microscope image (10 ×, NA = 0.25). The minimum size of a resolvable gap is decreased by a factor of 2.5 owing to SM-OCT. Note that the visibility of the gap is limited only by the speckle created by the turbid PDMS-$TiO_2$ phantom.

An approximation of the local phase variation introduced by the diffuser is $\phi_{\text{diffuser}}(x_d + x, y_d + y) = k\Delta nd(x_d + x, y_d + y)$, in which $k$ is the average wavenumber of the OCT illumination (note that a full description of the phase would require a vectorial electromagnetic simulation). The OCT beam with this phase variation is then imaged from the diffuser plane onto the sample by a $4f$ imaging system that smooths the beam's spatial features by a point-spread function, $PSF_{4f}$ (defined by lens $L_2$ in Fig. 1b). The beam then propagates back from the sample through the lenses and the diffuser (Supplementary Fig. 3). The OCT signal intensity is calculated as the interference between the field from the sample arm and a reference field. This simulation helps predict how different diffusers will perform for SM-OCT and provides insight into why certain diffusers cancel speckle while others exhibit worse performance. Supplementary Note 2 shows the application of the above model to the optical setup described in this manuscript along with simulations of the diffusers used in this study.

We characterized the thickness profiles of the diffusers used in this study (Supplementary Fig. 4), as well as their effects on the power on the sample (Supplementary Fig. 5 and Supplementary Table 1), signal intensity (Supplementary Fig. 5 and Supplementary Table 2) and lateral resolution (Supplementary Fig. 6 and Supplementary Table 3). Detailed acquisition parameters for all presented images are provided in Supplementary Table 4. OCT and SM-OCT images in this study are depicted on a logarithmic scale, with black and white representing low and high signal intensities, respectively.

**Results and analysis on phantoms.** A key advantage of SM-OCT is its ability to enhance visibility so that closely spaced scattering objects can be distinguished, therefore improving the effective resolution of the instrument by effectively eliminating speckle noise. Improvement in lateral resolution has also been observed in digital holography when using a ground glass diffuser to remove speckle[27]. Unlike holography, our study shows resolution improvement within a densely scattering sample, which creates speckle by backscattering light from multiple locations within the imaged voxel. We define the effective resolution as the width of the minimum detectable gap in a phantom, as detected and measured by an image segmentation algorithm. This gap corresponds to the smallest possible feature that is detectable within a scattering sample, such as tissue. Note that, unlike other common resolution criteria, our resolution definition specifically accounts for image noise, which is dominated by speckle in OCT. We demonstrated and quantified the improvement in effective resolution by imaging a small gap in a phantom made of titanium dioxide ($TiO_2$) powder dispersed in polydimethylsiloxane (PDMS). A narrowing gap was created by adjoining two rectangular pieces of the phantom at an angle (Supplementary Fig. 7). Using SM-OCT, we were able to detect a gap that was 2.5 times smaller than the smallest gap detected with OCT as determined by image segmentation (Fig. 2, Supplementary Note 3 and Supplementary Figs 7–9). Measurements of a resolution test target (Supplementary Fig. 6 and Supplementary Table 3) showed that the smallest resolvable separation in a non-turbid sample (7.5 μm in OCT and 12.2 μm in SM-OCT) was smaller than the smallest gap measured on the phantom (31.5 μm in OCT and 14 μm in SM-OCT). These results prove that speckle noise effectively limits feature visibility in most OCT images and that SM-OCT is able to recover the loss in effective resolution.

OCT speckle noise follows a Rayleigh distribution[15,33] (as a note, the local speckle contrast may follow a different distribution[34]). Thus, speckle noise can be considered to be eliminated when the pixel statistics are governed by scatterer distribution statistics rather than speckle statistics. We experimentally demonstrated this change in statistics by measuring the pixel value distribution within a phantom made of gold nanospheres (GNSs, 100 nm diameter, Sigma-Aldrich) dispersed in an agarose gel, as imaged with OCT (Fig. 3a) and SM-OCT (Fig. 3b). Owing to the strong backscattering and high concentration of the metallic nanoparticles, the agarose–GNS phantom is an excellent model for turbid media, and it produced Rayleigh speckle statistics as expected for conventional OCT imaging (Fig. 3c). In contrast, the pixel value distribution obtained with SM-OCT (Fig. 3d) became narrower with increasing averages (as predicted by the simulation in Supplementary Note 1 and Supplementary Fig. 1) and resembled a Poisson distribution, the expected pixel value distribution for scatterers randomly dispersed within a phantom[35] (see Supplementary Note 4 for a mathematical description). Additional sources for signal variability in the SM-OCT image are the absorption of the sample, size variability of the scattering nanoparticles, distance from the focal plane and residual illumination variability, which was created by the

diffuser and is characterized in Supplementary Note 5 and Supplementary Figs 10–12.

To further validate that SM-OCT removes speckle noise, we compared experimental data to the theoretical decrease of speckle contrast, proportional to $1/\sqrt{M}$, where $M$ is the number of compounded images with uncorrelated speckle noise[11]. We define $C$, the normalized s.d., as the s.d. of pixel intensities, which includes signal variations due to speckle noise, the sample and the imaging system, divided by the average pixel intensity in the same region (Supplementary equation (13)). Conventional OCT images exhibited minimal reduction (5.5%) in $C$ even with extensive averaging ($M = 100$), indicating that speckle noise was not affected by averaging. By comparison, SM-OCT imaging with equivalent averaging led to a significant reduction (77.8%) in $C$ as a result of the reduction in speckle noise (Fig. 3e). Since $C$ is composed of both speckle noise and the intrinsic random distribution of particles in the phantom, we define the normalized speckle (see Supplementary Note 6), which decreases by a factor of $\sqrt{M}$ theoretically (Supplementary equation (16)) and experimentally (Fig. 3f). The demonstration that the reduction of speckle noise is inversely proportional to $\sqrt{M}$, as expected from speckle theory, indicates that SM-OCT does indeed obtain $M$-independent speckle patterns in $M$-acquired frames, for at least 100 frames.

The speckle reduction achieved with SM-OCT reveals fine structures that are typically obscured by noise. As a practical demonstration of this ability, we embedded large gold nanorods (LGNRs, 30 nm wide and 100 nm long)[36] and polystyrene beads of 3 μm diameter inside an agarose phantom (Fig. 4a). As predicted, speckle noise was predominant in conventional OCT images and consequently most of the beads were not visible (Fig. 4b,d,f). Conversely, SM-OCT enabled detection of the beads in the presence of the random signal originating from the LGNRs, which is influenced by their random positions and orientations (Fig. 4c,e,g). The evolution of the images as the number of averages increases (Fig. 4f,g) showed that the beads were more easily detected in the SM-OCT image compared to OCT after as few as 10 averages. As the number of averages increased, the bead locations (Fig. 4h) became more visible in SM-OCT, while the OCT image remained obscured by an unchanged speckle pattern. Note that when the number of averaged images was low, photon shot noise was significant. The signal intensity profiles (Fig. 4i,j) show the reduction of speckle noise and the presence of the beads identified using SM-OCT, while in the OCT profiles some of the beads were not visible or were indistinguishable from the intensity of speckle noise. Images of the phantom acquired with bright-field microscopy (Supplementary Fig. 13a) and SM-OCT revealed a sparse distribution of beads inside the agarose-LGNR phantom. This comparison indicated that SM-OCT yielded a more accurate representation of the structure of the sample than OCT. Imaging an agarose–$TiO_2$ nanopowder phantom containing $TiO_2$ aggregates further validated the capability of SM-OCT to produce images that better represent the true structure of the sample (Supplementary Fig. 13b–d).

**Results in living intact tissue.** One of the greatest biomedical advantages of OCT is its ability to provide non-invasive high-resolution images of intact living tissues. However, strong speckle artefacts drastically limit the ability to resolve fine anatomical structures. By removing the significant contribution of speckle noise, SM-OCT is capable of rendering *in vivo* images that approach histological detail. Figure 5 depicts OCT and SM-OCT images of a mouse ear pinna, which consists of well-defined epithelial and cartilage layers, small blood and lymph vessels,

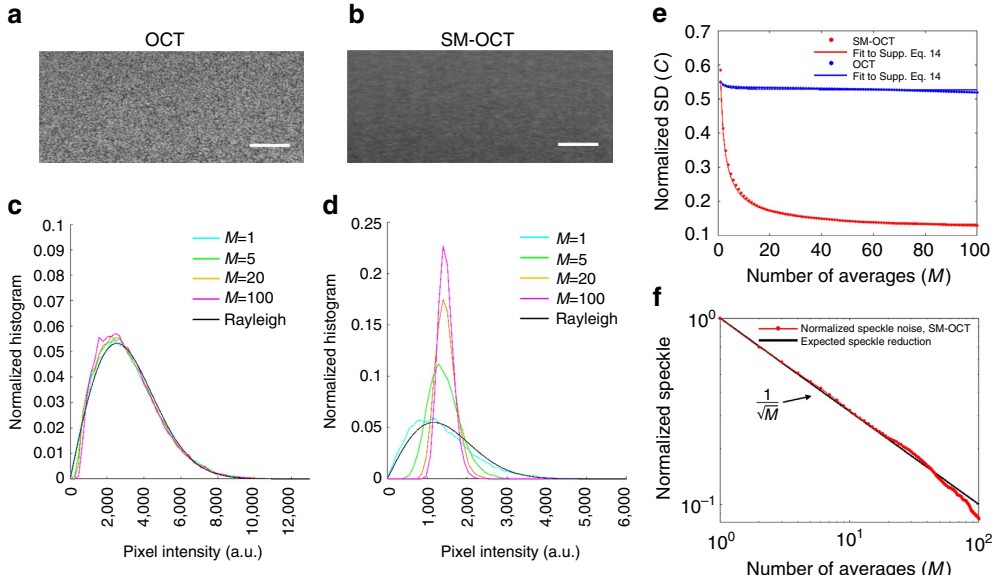

**Figure 3 | Analysis of speckle statistics and speckle contrast in SM-OCT and conventional OCT.** (**a,b**) OCT and SM-OCT images of GNSs dispersed in agarose. The OCT image shows a combination of speckle noise and the signal variation from the random distribution of GNSs in the phantom. The SM-OCT image shows only the latter. This claim is supported by the statistical analysis of pixel intensities. Scale bar, 100 μm. (**c,d**) Statistical analysis of the pixel values shows that the OCT image (**c**) is dominated by speckle noise and the distribution of pixel values is approximately a Rayleigh distribution that persists with averaging ($M$ is number of averages). In SM-OCT (**d**), increasing the number of averages narrows the distribution significantly. (**e**) Reduction in normalized s.d. versus the number of averages, $M$, for OCT and SM-OCT. The reduction in the normalized s.d. is significantly larger in SM-OCT versus OCT. (**f**) The reduction of normalized speckle as defined by Supplementary equation (14) (see Supplementary Note 6) follows $1/\sqrt{M}$, as expected.

and numerous hair follicles and sebaceous glands. Many of these structures were masked by speckle noise in OCT but became visible in SM-OCT images. Speckle removal revealed fine structures in cross-sectional B-scans (Fig. 5a–f and Supplementary Movies 1–3) as well as in frontal (*en face*) sections (Fig. 5g,h), indicating that SM-OCT provided major improvement in image quality in all three spatial dimensions. Figure 5f shows a 2 μm-thick horizontal line, demonstrating that the intrinsic axial resolution (defined by the spectral bandwidth of the OCT) is uncompromised. The *en face* SM-OCT image shows lymph vessels and fine structures that are more visible compared to the OCT image (Fig. 5g,h). Histological sections of the pinna (Fig. 5i) show the small structures that were also observed clearly in SM-OCT images but not in OCT images. These features can be delineated in volumetric SM-OCT renderings (Fig. 5j and Supplementary Movie 3). We further compared SM-OCT images with alternative speckle reduction methods (Supplementary Note 7 and Supplementary Figs 14–16). SM-OCT outperformed each of these methods with respect to speckle noise reduction. Moreover, noise suppression with these methods came at the cost of smoothing fine features, while SM-OCT yielded images with uncompromised structural detail.

As a second *in vivo* demonstration of SM-OCT, we acquired images of the cornea and retina of a live mouse. Using SM-OCT, we were able to see the lamellar structure of the corneal stroma as well as clear boundaries between various layers of the cornea (Fig. 6a–e). Owing to speckle noise, conventional OCT was unable to resolve these features. We then imaged the retina of a live mouse[37]. The individual layers of the retina were particularly well resolved with SM-OCT (Fig. 6f–i). For example, the outer plexiform layer and the external limiting membrane can be readily distinguished in SM-OCT images.

To enable more robust SM-OCT imaging of moving samples, we implemented our approach using A-scan averages instead of frame (B-scan) averages. This approach required moving the diffuser fast enough such that it is sufficiently translated between A-scan acquisitions, thereby resulting in uncorrelated speckle patterns that form a virtually speckle-free A-scan when averaged. For this purpose, the diffuser was moved rapidly and continuously using a rotating mount, which provided uncorrelated speckle patterns in each A-scan (Supplementary Note 8 and Supplementary Fig. 17). We used this setup to image the cornea of a mouse *in vivo* (Supplementary Fig. 18). Note that moving the diffuser too fast can induce multiple phase changes during the acquisition time of a given A-scan, which will result in OCT fringe washout and a decrease in signal intensity.

To demonstrate the potential of SM-OCT in dermatological applications, we imaged the fingertip skin of a human volunteer (Fig. 7, Supplementary Figs 19 and 20 and Supplementary Movie 4). The speckle noise reduction achieved with SM-OCT enabled detection of fine structures including sweat ducts, dermal papillae and tactile corpuscles (Meissner's corpuscles). To our knowledge, this demonstration is the first time that the tactile corpuscle has been clearly observed in the intact skin of a live human. SM-OCT was particularly helpful in identifying the boundaries between the corpuscle and the surrounding dermis. As in images of the mouse cornea, SM-OCT images of the fingertip revealed the cellular structure and striation of the tactile corpuscle, proving that SM-OCT can remove speckle noise without compromising resolution. This example suggests that SM-OCT may be used to improve non-invasive dermatological studies in humans by producing images that approach the quality of histology.

We also performed SM-OCT retinal imaging of a human volunteer (one of the authors). Supplementary Fig. 21 depicts images of the human retina obtained with SM-OCT using the retinal OCT system. Optical removal of speckle resulted in enhanced delineation of the various retina layers, seen most clearly in all three nuclear layers as well as in the differentiation between the outer retina layers.

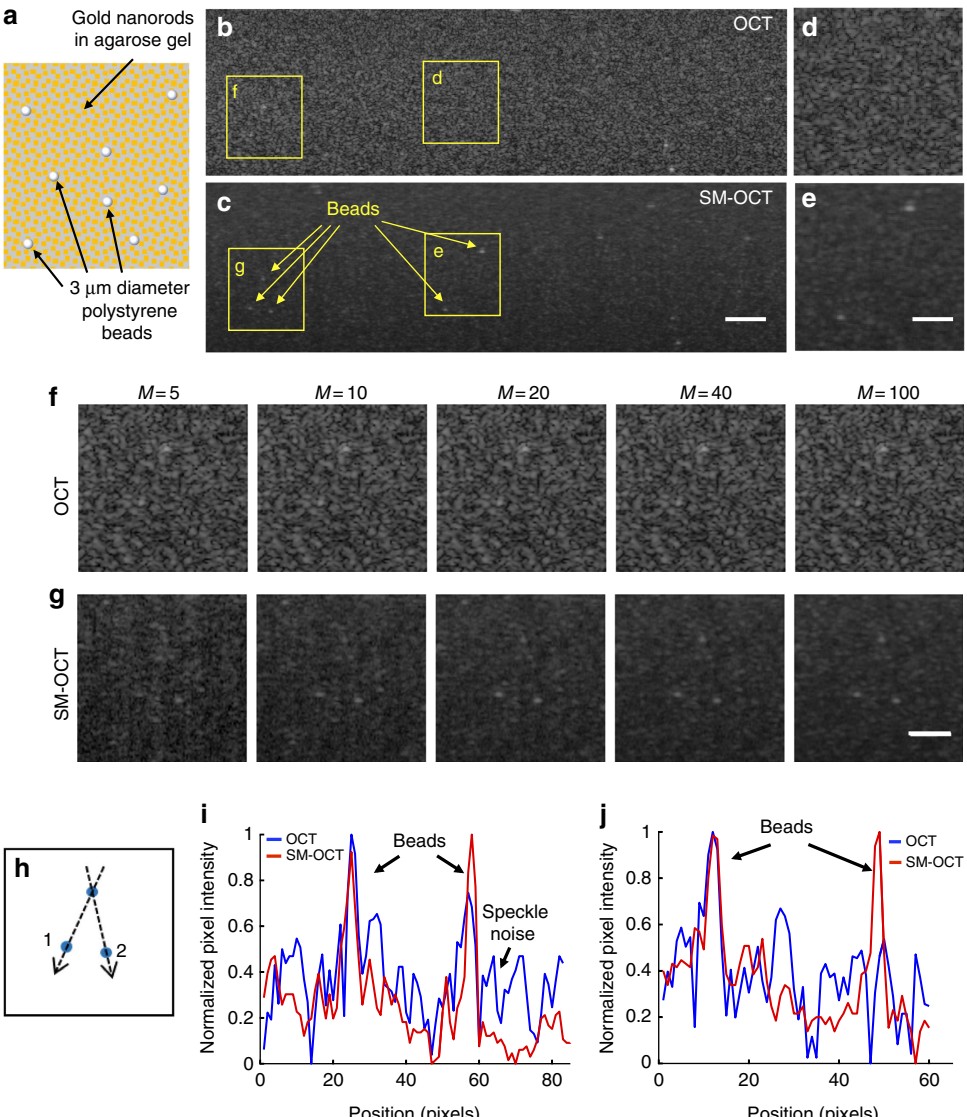

**Figure 4 | Demonstration of speckle removal and increased visibility in phantoms.** (**a**) Schematic of a phantom made by dispersing LGNRs and 3 μm diameter polystyrene beads in an agarose gel. (**b,c**) OCT and SM-OCT B-scans of the phantom. In the OCT image, many of the beads cannot be detected due to speckle noise. In the SM-OCT image, speckle noise is significantly reduced while preserving resolution, and the beads become visible, along with the random distribution of LGNRs in the phantom. Scale bar, 100 μm. (**d–g**) Close-up views of regions in the phantom showing superiority of SM-OCT over OCT in detecting the beads. In the SM-OCT image the beads are revealed as the number of averaged images (M) increases. Scale bar, 50 μm. (**h**) Schematic showing the locations of the three beads. (**i,j**) Intensity profiles (on logarithmic scale) along lines 1 and 2, respectively, as depicted in **h**, demonstrating the beads are easily visible in SM-OCT but not in OCT. Size of a pixel is 2 μm.

## Discussion

We have demonstrated SM-OCT, a technique that is able to efficiently reduce speckle noise arbitrarily well in OCT by utilizing a moving diffuser to locally induce random phase shifts in the light illuminating and collected from each voxel. In addition to being highly effective for speckle removal, SM-OCT is a low-cost, robust and simple modification to existing OCT systems. In this study, SM-OCT was integrated as an extension to two commercial OCTs with basic components. Our implementation utilized a ground glass moving diffuser to reduce speckle; however, the same physical effect can likely be achieved by other means (for example, a spatial light modulator), provided that phase changes are introduced within each voxel. The ability to scramble the phase inside the voxel is limited by the PSF of the lenses in the 4f imaging system. Therefore, these lenses should have a smaller PSF than that of the main lens of the OCT, which

defines the system's lateral resolution and voxel size. This means that the size of the acquired voxel in OCT must be deliberately larger in order to introduce phase scrambling within it. However, the benefit of speckle noise removal significantly outweighs the reduction in lateral resolution because speckle removal ultimately allows detection of fine structural details.

This study presents the first implementation of a moving diffuser in OCT and the first demonstration of practically speckle-free high-resolution three-dimensional (3D) volumes of turbid media and living tissue. Although the use of moving diffusers has been demonstrated for speckle reduction in microscopy[22] and holography[26], these are not able to obtain volumetric images of densely scattering samples, such as tissue. Thus, these methods are often limited to characterization of single layers of cells or reflective samples, such as semiconductor wafers. The implementation of SM-OCT is also distinct from these prior

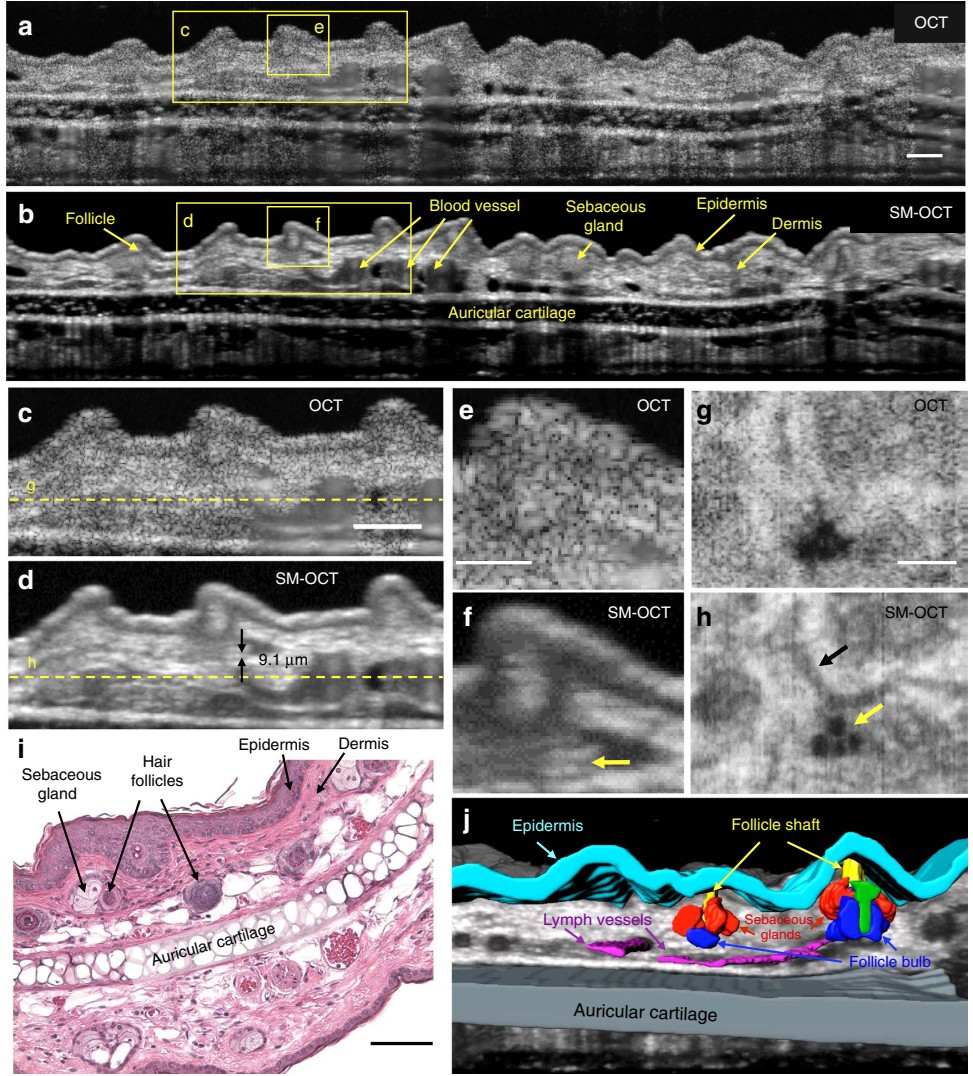

**Figure 5 | SM-OCT in the mouse pinna reveals fine low-contrast structures.** (**a**,**b**) OCT and SM-OCT B-scans of a mouse pinna. Scale bar, 100 μm. (**c**,**d**) Close-up views on the regions marked in **a**,**b**. The arrows in **d** depict an anatomical feature the size of 9.1 μm that is not visible in the OCT image. Scale bar, 100 μm. (**e**,**f**) Close-up views on the regions marked in **a**,**b**. The arrow in **f** shows a dark line, which is 2 μm thick. Scale bar, 50 μm. (**g**,**h**) OCT and SM-OCT *en face* scans at the depth indicated by the dashed line in **c**,**d**. The SM-OCT image in **h** shows lymph vessels (black arrow) and fine structures (yellow arrow) that are nearly invisible in **g**. Scale bar, 200 μm. (**i**) Microscope image of a haematoxylin and eosin (H&E)-stained mouse ear pinna at 10 × magnification. Scale bar, 100 μm. (**j**) Manual segmentation of the ear volume is possible owing to the removal of speckle noise, revealing the structure of hair follicles (Supplementary Movie 3). Cyan—epidermis, grey—auricular cartilage, magenta—lymph vessels, red—sebaceous glands, blue—follicle bulb, yellow—follicle shaft, green—unidentified part of the follicle.

methods. In SM-OCT, the diffuser is placed within the OCT's sample arm and moved in a conjugate image plane created by a 4*f* imaging system. In order to achieve sufficient speckle decorrelation, the diffuser must have certain roughness features, as discussed previously. Random laser illumination[38] and low-spatial-coherence semiconductor lasers[39] have been proposed as speckle-free light sources for imaging; however, they have not been demonstrated to produce tomograms. Furthermore, SM-OCT is able to reduce a potentially unlimited amount of speckle originating from the turbid sample itself (caused by multiple backscattering from the imaged voxel) in addition to speckle caused by a turbid object placed in the optical path (caused by multiple forward scattering).

The inverse proportion of speckle noise to $\sqrt{M}$ has been previously demonstrated for OCT[13,40]; however, these compounding methods were limited in the number of independent speckle patterns they could achieve. In contrast to these methods, SM-OCT is shown here to obtain an unprecedented number of 100 independent speckle patterns (Fig. 3f), which could be extended without compromising resolution, thereby exceeding the limitations of previously described OCT compounding methods, and enabling detection of small structures with remarkable clarity.

As is true for most compounding methods, SM-OCT requires averaging of several OCT images that, in the current implementation, extends the time of image acquisition. One theoretical limitation of our method is that an object cannot move more than a few microns while frame averages are acquired. However, this requirement is merely an artefact of acquiring frame (B-scan) averages instead of A-scan averages, which were demonstrated in Supplementary Fig. 18 by using a rotating diffuser. In this case the diffuser should move fast enough to create uncorrelated speckle patterns in every A-scan, but slow enough to avoid washout of the interference fringes. If an OCT with a very fast A-scan rate is

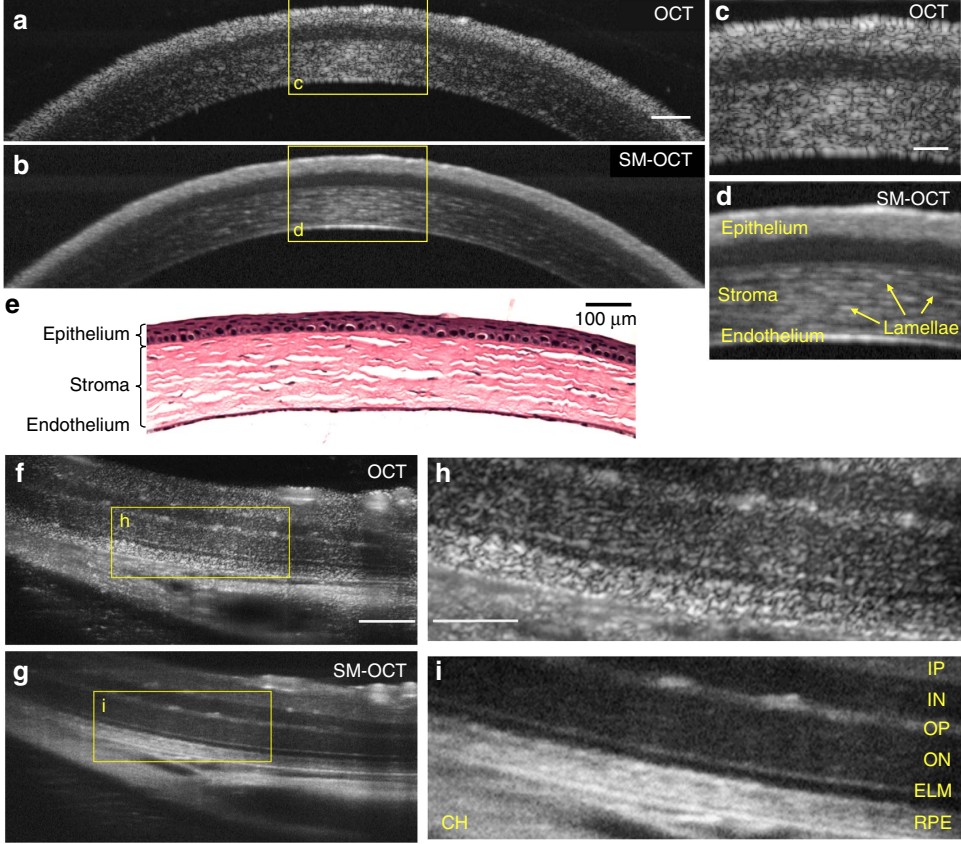

**Figure 6 | SM-OCT imaging of the mouse cornea and retina clarifies the boundaries between the layers and reveals the cellular structure of the stroma.** (**a,b**) OCT and SM-OCT B-scans of a mouse cornea. Scale bar, 100 μm. (**c,d**) Close-up view on the regions marked in **a,b**. In **c**, due to the high density of scatterers in this tissue, speckle noise is masking the inner structure of the stroma. Scale bar, 50 μm. (**e**) Microscope image of H&E-stained mouse cornea at 10 × magnification. Scale bar, 100 μm. (**f,g**) OCT and SM-OCT B-scans of a mouse retina. Scale bar, 100 μm. (**h,i**) Close-up view on the regions marked in **f,g**. IP, inner plexiform; IN, inner nuclear layer; OP, outer plexiform layer; ON, outer nuclear layer; ELM, external limiting membrane; RPE, retinal pigment epithelium; CH, choroid. Scale bar, 50 μm.

used, the rotation speed of the diffuser can be increased by using a faster motor or a larger diffuser, since the tangential velocity is proportional to the diffuser radius. Another way to acquire SM-OCT images of fast-moving objects is by implementing a conventional tissue-tracking system[41] or a system that can achieve image compounding without extending the acquisition time, such as interleaved OCT[42]. Overall, even with the current implementation, we do not expect the averaging requirement to limit SM-OCT imaging because significant speckle reduction can be achieved with as few as 10 averages, an amount already used in conventional OCT to reduce photon and thermal noise. Further, we demonstrate here that, despite the increase in acquisition time, it is possible to image living subjects' skin and eyes. As hardware advances continue to increase OCT acquisition rates, SM-OCT acquisition times will reduce concurrently. A detailed discussion of the effects of SM-OCT on resolution and signal intensity can be found in the Supplementary Discussion.

The implementation of SM-OCT described in this manuscript is best suited for SDOCT systems, in which the full spectrum is captured simultaneously. For time-domain OCT and swept-source OCT, the continuous movement of the diffuser during acquisition may pose challenges because of fringe washout; thus, it may be beneficial to synchronize the movement of the diffuser so that it moves only in between frame or A-scan acquisitions. Another challenge of SM-OCT is that phase and speckle variance methods, such as those used in Doppler OCT and OCT angiography[8], will encounter additional variations owing to the movement of the diffuser. In order to obtain a high-resolution speckle-reduced volume and angiography of the same volume, the sample should be scanned once with a moving diffuser and a second time with the diffuser either static or removed from the optical path.

In summary, we expect that SM-OCT will enable superior diagnostic capabilities compared to conventional OCT because of its ability to reveal anatomical features that are otherwise hidden by speckle noise. Potential clinical applications of SM-OCT include early detection of epithelial cancers, evaluation of tumour margins, early detection of retinal diseases and internal diagnostics (see Supplementary Fig. 22 for a proposed design of an SM-OCT endoscope). In addition, the significant reduction of speckle noise facilitates further OCT image enhancement and image-based calculations including measurement of the attenuation coefficient[10,43] (Supplementary Fig. 23 and Supplementary Note 9), blur-deconvolution for extended depth of field and improved segmentation of structures such as retinal layers[44,45], which will aid the early diagnosis of diseases.

## Methods

**Experimental setup.** SM-OCT was implemented by modifying two existing OCT systems: the Ganymede HR (Thorlabs) and a clinical retinal imaging device (iFusion, Optovue). Both are SDOCT systems. All SM-OCT images except for the human retina images were acquired using the Ganymede HR (HR-OCT).

The implementation of SM-OCT on the Ganymede HR appears in Fig. 1. The light source of the Ganymede HR is a super luminescent diode with a centre wavelength of 900 nm. The spectrometer has a 200 nm bandwidth ($\lambda = 800$–$1,000$ nm), which provides 2.1 μm axial resolution in water. The spectrometer acquires 2,048 samples for each A-scan at a measured rate of

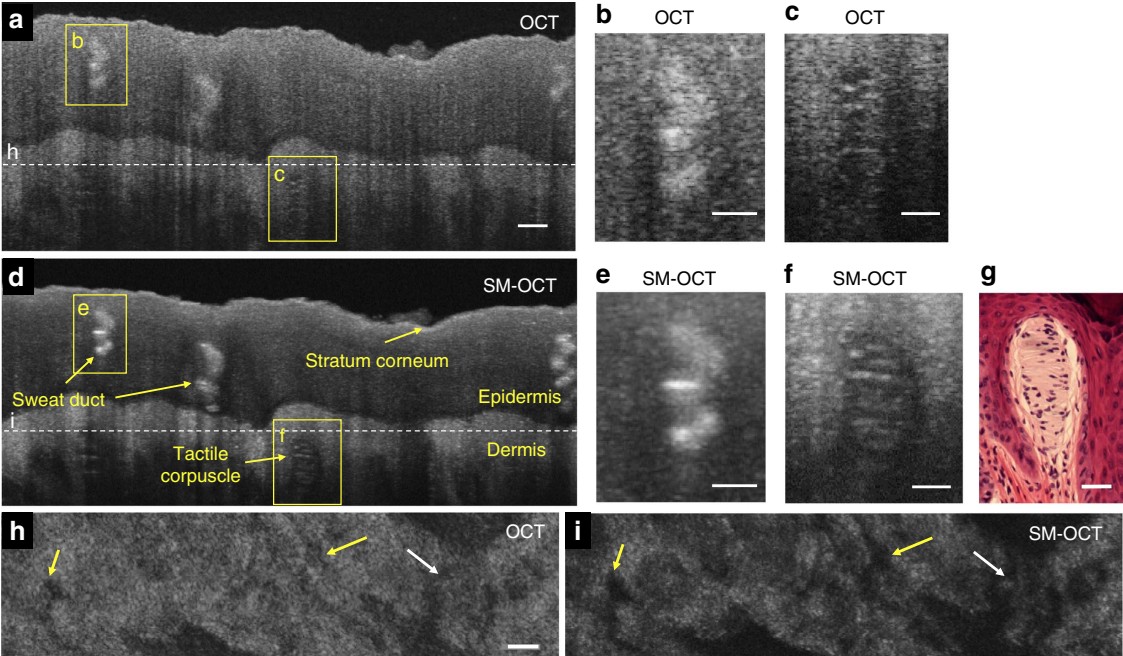

**Figure 7 | SM-OCT imaging of intact human fingertip skin reveals fine structures such as the tactile corpuscle.** (**a**) OCT B-scan of a fingertip. Scale bar, 100 μm. (**b**) Close-up view on the sweat duct marked in **a**. Scale bar, 50 μm. (**c**) Close-up view on the tactile corpuscle marked in **a**. Scale bar, 50 μm. (**d**) SM-OCT scan of a fingertip. (**e**) Close-up view on the sweat duct marked in **d**. Scale bar, 50 μm. (**f**) Close-up view on the tactile corpuscle marked in **d**. Scale bar, 50 μm. (**g**) Microscope image of H&E-stained tactile corpuscle (courtesy of Dr Jesus Lozano, Dr Lorena Monarrez and Professor Doug Schmucker, Department of Anatomy, UCSF School of Medicine). Scale bar, 50 μm. (**h,i**) OCT and SM-OCT *en face* images from a 3D scan of the fingertip, located at the top of the dermis, as shown by the dashed line in **a,d**. With SM-OCT, there is an improved delineation of the dermal papillae (yellow arrows) and sweat ducts (white arrow). Scale bar, 100 μm.

20.7 kHz. All image reconstruction and analyses were performed with Matlab (Mathworks) using raw data from the spectrometer. The first lens of the imaging system (LSM03-BB, Thorlabs) provides a lateral resolution of 8 μm (full-width at half-maximum, FWHM) and depth of field of 143 μm in water. In the Ganymede HR, the diffuser was placed at the original focal plane of the OCT probe, and a new focal plane was projected by a 4f imaging system[46]. The 4f configuration was implemented using two similar lenses (LSM02-BB, Thorlabs) that provide a lateral resolution of 4.2 μm (FWHM) and depth of field of 32 μm in water. Owing to the extension of the sample arm and the addition of two lenses and the diffuser, the reference arm was extended by ~10 cm, and dispersion compensation elements were added (two LSM02DC, Thorlabs). The reference arm was extended by placing metal rods between the OCT probe and the reference mirror. OCT images were obtained with the SM-OCT apparatus without the diffuser. In this configuration, light propagates through the 4f imaging system and the extended reference arm. OCT images obtained this way are of similar quality to the OCT images obtained with the original probe. The only difference from the original probe is a 9% loss of power on the sample (Supplementary Table 1), which reduces signal-to-noise but does not change the properties of speckle. The diffuser was placed in the focal plane of the first lens and held within a custom-motorized mount with XYZ translation (based on CXYZ1, Thorlabs). The diffusers were moved by a motor (Z812, Thorlabs), back and forth along one axis and controlled through computer software (APT, Thorlabs). The movement of the diffuser was always perpendicular to the direction of the B-scan. The diffuser was translated back and forth at 0.3 mm s$^{-1}$ over a range of 6.5 mm and an acceleration of 1.5 mm s$^{-2}$. Change in the direction of the diffuser occurred during the scan only when acquiring large volumes. Such volumes were acquired three times and the three acquisitions were averaged to obtain a volume in which the effect of the moving diffuser was observed throughout the volume. For the implementation of A-scan-based SM-OCT using a fast-rotating diffuser, the diffuser was placed in a rotating motor (RSC-103, Pacific Laser Equipment). The OCT beam was focused near the outer edge of the diffuser where the velocity was ~9 mm s$^{-1}$. This velocity was sufficient to create decorrelated speckle patterns using our OCT system, which has a measured A-scan rate of ~20 kHz (Supplementary Figs 17 and 18). Therefore, in this setup, the number of uncorrelated speckle patterns is equal to the number of acquired A-scans.

The implementation of SM-OCT on the retinal system appears in Supplementary Fig. 2. The iFusion is based on the iVue SDOCT. The scan beam centre wavelength is $\lambda = 840 \pm 10$ nm and provides an axial and lateral resolution of 5 and 15 μm in the retina, respectively. Each frame is composed of 1,024 A-scans that are acquired at 26 kHz. The images in this study (Supplementary Fig. 21) were acquired in Retina Cross Line Mode with 44 B-scan averages and a software

parameter set to include all frames in the average. The diffuser was placed in the conjugate image plane. In the retinal system, it was not necessary to project a new focal plane because one such plane is accessible inside the original OCT probe. Hence, the retinal implementation of SM-OCT is simpler, as the change in the sample arm is negligible and, therefore, does not require extension of the reference arm. A dichroic mirror, which is used for obtaining fundus images, was removed to make room for the diffuser. The diffuser was held within a thin fixed mount (LH-1T, Newport) that was attached to a motorized translation stage. The stage and the diffuser were moved along one axis, perpendicular to the direction of the B-scan, and controlled through computer software (APT, Thorlabs). The diffuser was translated at a speed of 1.5 mm s$^{-1}$ over a range of 6 mm. All processings were done internally with the iFusion computer and software.

The diffusers used for all experiments are ground glass diffusers with antireflective coating on one side (DG–1500-B and DG10-2000-B, Thorlabs). The 1,500 and 2,000 grit diffusers are 2 and 1 mm thick, respectively. The 3 μm lapped diffuser was created by further lapping a commercial 1,500 grit diffuser with 3 μm aluminium oxide grit (Universal Photonics) for 15 min. The profile and height statistics of the diffusers appear in Supplementary Fig. 4.

Accurately positioning the diffusive plane of the diffuser at the waist of the Gaussian beam of the OCT was crucial for obtaining high-quality images. Deviations from the ideal diffuser axial location resulted in power and resolution losses and impeded the speckle reduction effect. We manually positioned the diffuser at this ideal location by changing the location of the diffuser along the optical axis until an optimal image was obtained. In the retinal OCT system, the optimal placement of the diffuser along the optical axis was found by a combination of two indications. First, the person being examined adjusted the location of the diffuser until an optical test target (inherent to the iFusion) appeared in sharp focus to the examinee. Next, the person acquiring the images adjusted the location of the diffuser to obtain an optimal signal-to-noise within the OCT image of the retina.

**Characterization of ground glass diffusers.** Intuitively, to achieve maximal phase decorrelation the random phases added by the diffuser, $\theta_{n,m}$ (equation (1)), should be evenly distributed between 0 to $2\pi$ at the OCT focal plane. In order to obtain this phase shift using a diffuser made of glass with a refractive index of 1.5 (NBK-7) and light sources with a centre wavelength of 900 nm, the total thickness variation of the diffuser should span at least 1.8 μm. However, deflection of light by the diffuser, which is more probable in a ground glass diffuser with a large thickness variation, reduces the OCT signal and should be minimized. In our implementation, we used three types of diffusers and characterized their thickness

and roughness (Supplementary Fig. 4) with a 3D optical profiler. The roughest diffuser is a commercial 1,500 grit diffuser with antireflective coating. The finest diffuser was made by further grinding (lapping) the 1,500 diffuser with 3 μm particles (3 μm lapped diffuser). We also used a 2,000 grit diffuser, which has a roughness between the previously mentioned diffusers. Each of the three diffusers tested had a small effect on the optical power on the sample (Supplementary Table 1), the OCT signal (Supplementary Table 2) and the lateral resolution (Supplementary Fig. 6 and Supplementary Table 3). While the roughest diffuser (1,500 grit) reduced the OCT signal and the lateral resolution more than the other diffusers, it achieved the best qualitative performance in terms of speckle removal and appearance of fine anatomic detail in tissue.

The profiles of the diffusers were measured with a non-contact 3D optical profiler (S neox, Sensofar). The profiles were obtained with a $50\times$ magnification objective lens (Nikon, numerical aperture (NA) 0.55 50X Nikon CF IC Epi Plan DI Interferometry Objective) in an interferometric scan mode. Post processing was performed with the SensoSCAN programme (Sensofar) and included depth slope correction and calculation of the depth histogram. In addition, the profile of the 1,500 grit diffuser required restoration (interpolation) due to regions from which light was not collected.

**Processing and display.** The post-processing methods in this section were applied only for the HR-OCT system (Ganymede HR, Thorlabs). Post processing was done with Matlab (2015a and 2014b, Mathworks). The raw spectrum of each A-scan acquired with OCT and SM-OCT was processed in a similar way to create images in the spatial domain. Reconstruction was performed by subtracting the spectrum of the source, as measured by the OCT, and multiplying by a phase matrix that is equivalent to applying a Fourier transform[47]. To reduce spectrum-derived artefacts, the spectrum was multiplied by a Hann window with 2,048 points. Prior to reconstruction, dispersion compensation was performed by finding the coefficient of the quadratic phase term iteratively by minimizing the absolute difference between the reconstructed images of two distinct spectral windows[48,49]. Dispersion compensation was done separately for each experiment. To obtain the final OCT/SM-OCT image, the magnitudes of multiple reconstructed B-scans were averaged on a linear scale.

In order to minimize movement artefacts, frames that were notably different from most of the frames in the scan were excluded from averaging. This exclusion process was done only for the mouse retina and mouse cornea images, in which there was significant movement due to breathing. Frame similarity was determined by measuring the correlation of each frame to the average of all the frames. The threshold for excluding frames was determined manually for each scan.

The number of averages for each image appears in Supplementary Table 4. The averaged image is displayed on a logarithmic scale with image-adaptive brightness scaling unless otherwise stated. Dark pixels correspond to low scattering from the sample, while bright pixels correspond to high intensity of scattering.

**Phantom preparation.** The PDMS-TiO$_2$ phantom (Fig. 1e) was fabricated by spin-coating layers of PDMS (Sylgard 184 Silicone Elastomer, Dow/Corning) comprising TiO$_2$ powder particles (TiO$_2$ anatase, 232033, Sigma-Aldrich) with an average size of $130 \pm 70$ nm.

Agarose phantoms embedded with various scattering agents were created using a stock solution of agarose (J.T. Baker) in water. Following three different scattering agents were used: TiO$_2$ anatase nanopowder with 21 nm primary particle size (Sigma-Aldrich), GNSs with 100 nm diameter (Sigma-Aldrich) and LGNRs with peak absorption at 745 nm and size of $\sim 90$ by 35 nm (ref. 48). LGNRs were used because their scattering-to-absorption ratio is higher compared to conventional gold nanorods[50] and because their scattering peak is in the infrared. The agarose–GNS phantom (Fig. 3) consisted of $10^{11}$ GNSs per ml (corresponding to 12 nanoparticles per voxel) in a 5% agarose solution. The higher concentration of agarose was needed to decrease the pore size of the agarose gel and eliminate the diffusion of the nanoparticles in the phantom[51]. The agarose-LGNR phantom with beads (Fig. 4) consisted of $2 \times 10^{11}$ LGNRs per ml (corresponding to 33.6 LGNRs per voxel) and polystyrene beads (Streptavidin Polystyrene Particles, average diameter 3.05 μm, 0.5% w/v, Spherotech) at a final concentration of $2.38 \times 10^8$ beads per ml (0.04 beads per voxel) in a 1% agarose solution. The agarose–TiO$_2$ phantom (Supplementary Fig. 13) was fabricated by dispersing 0.009 g of nanopowder in 1 ml ultrapure water. The solution was sonicated; however, the clumps persisted. For the three phantoms described above, the scattering agents and polystyrene beads were slowly added to 5 ml of uncured 1% agarose solution at 60 °C with continuous stirring. The final solution was allowed to stir for 1 min before being poured into 5 ml plastic Petri dishes. The phantoms were allowed to cure for at least 2 h before imaging.

**Optical power and signal intensity.** The optical power on the sample and the OCT signal was measured for OCT and SM-OCT with the three different diffusers on the HR-OCT. The optical power was measured by placing a power meter (PM122D, Thorlabs) with a germanium sensor (S122C, Thorlabs) and aperture 9.5 mm at the focal plane of the scan lens while scanning at a single point at the centre of the field of view. The measurement was calibrated for the centre wavelength of the source, 900 nm. At least 100 consecutive measurements were

acquired with the power meter for a time period of $\sim 60$ s. The OCT measurement refers to the original probe without any additional components. The measurement named 'no diffuser' refers to the SM-OCT system, which adds two lenses to the original probe, without a diffuser. The signal intensity was measured on images of a PDMS + TiO$_2$ phantom with 100 B-scan averages. The regions (500 μm long and 100 μm deep) selected for the measurements were all chosen at the same depth in the phantom, location relative to the focal plane and position on the screen, to eliminate the effect of absorption, focusing and signal roll-off. The values are on a linear scale and in arbitrary units. The relatively high s.d. in the signal intensity is due to aggregations of TiO$_2$, distance from the focal plane and absorbance inside of the region selected for this measurement. Note that the decrease in signal intensity due to the diffusers is larger than the decrease in power on the sample, because the signal is created by light that is travelling twice through the diffuser and because some light that was measured by the power meter at the sample will be rejected via the confocal detection of our OCT system.

**Measurement of lateral resolution.** The lateral resolution of OCT and SM-OCT was evaluated using a 1951 USAF glass slide resolution target (Edmund Optics). The target was placed at the focal plane and scanned in a 3D mode three times, each with 18 B-scan averages to obtain a total of 54 B-scan averages. The samples were scanned with 3 μm spacing in both lateral directions over a 1 by 1 mm square. The reconstructed volumes were averaged on a linear scale to create a single volume for each condition. Next, 200 rows that include the surface of the resolution chart were averaged along the depth axis to obtain an *en face* projection of the volume. The images are displayed on a logarithmic scale (Supplementary Fig. 6).

For each type of scan, the smallest resolvable group was determined visually in both horizontal and vertical directions. The inverse of the line-pairs-per-mm of the smallest resolvable group was used to calculate the effective FWHM of the beam. The PSF size increase is calculated as $1 - ((\text{SM-OCT average lines-pairs-per-mm})/(\text{OCT average lines-pairs-per-mm}))$. Note that the FWHM of the OCT, as defined by the scanning lens, is 11.2 μm in air. The effective resolution is better because we are determining the visibility of line separations visually on logarithmic scale images.

**Measurement of gap in phantom.** Images of the PDMS-TiO$_2$ phantom were acquired with a bright-field microscope ($10\times$, NA = 0.25); OCT and SM-OCT were registered and segmented to detect the gap in the phantom. See Supplementary Note 3 for detail regarding the registration of the images and the measurement of the gap.

**Imaging of live mouse.** All animal experiments were performed in compliance with the the Institutional Animal Care and Use Committee (IACUC) guidelines and with the Stanford University Animal Studies Committee's Guidelines for the Care and Use of Research Animals. Experimental protocols were approved by Stanford University's Animal Studies Committee. *Foxn1*$^{nu/nu}$ mice (Charles River Labs) were anaesthetized by inhalation of 2.5% isoflurane in O$_2$ (v/v). Once adequately anaesthetized, the right ear pinna was immobilized using double-sided tape. We optimized light transmission to the sample by applying ultrasound gel to the mouse skin and covering the gel with a 2 mm-thick one-sided antireflective-coated glass (650–1,050 nm), with the coating at the air–glass interface. For retinal and corneal imaging studies, mice were anaesthetized with intraperitoneal injections of 80 mg kg$^{-1}$ ketamine (Vedco Inc.) and 10 mg kg$^{-1}$ xylazine (Lloyd Inc.). Once adequately anaesthetized, the mice were mounted on to a platform and secured with a stereotactic device. With the mouse secured, the stage was tilted to orient the mouse's eye upwards, with the top of the cornea being approximately parallel to the table. Pupillary dilation was achieved by applying one drop each of 1% tropicamide (Bausch & Lomb), and 2.5% phenylephrine hydrochloride (Paragon BioTeck) to the eyes, for 2 min each. Hypromellose solution (2.5%; Gonak, Akorn Inc.) was then placed over each eye as a contact solution. Anaesthesia was continually maintained using a nose-cone delivering 1.5–2% isoflurane in O$_2$.

**Imaging of human fingertip.** The fingertips of healthy volunteers were imaged with the Ganymede HR. The subject's finger was pressed onto the bottom of a fixed glass window with antireflective coating at the air–glass interface. As in the setup for imaging the mouse pinna, we optimized light transmission to the sample by applying ultrasound gel to the fingertip skin. To minimize movement, SM-OCT scans were acquired in two-dimensional or 3D modes with the diffuser, and then the diffuser was quickly removed to acquire the corresponding conventional OCT images.

**Imaging of human retina.** A healthy human volunteer (one of the authors) was scanned with the retinal system (iFusion, Optovue). The Stanford Office of Human Subjects Research determined that this experiment did not warrant review by an institutional review board. The diffuser was placed in the image plane for SM-OCT imaging and removed for OCT imaging. Images were acquired in quick succession to reduce movement between scans.

**Supplementary movies.** OCT and SM-OCT volumes were made into tiff stacks. Supplementary Movie 1 was made with ImageJ[52]. Supplementary Movies 2–4 were made using Imaris (Bitplane). In Supplementary Movie 3, segmentation was done manually frame by frame using Matlab (Mathworks). The segmented volumes were combined with the original SM-OCT volume in Imaris.

**Measurement of exponential coefficients.** The attenuation coefficient of a sample may be calculated by fitting the OCT or SM-OCT signal intensity to a function that includes the effects of Beer–Lambert law (an exponential function), the confocal function, OCT roll-off and multiple scattering[27,53]. We expect that SM-OCT would enable a more precise fit owing to the removal of speckle noise. In order to compare the precision of the fit between OCT and SM-OCT, we performed an exponential fit to an image of a fingertip (Supplementary Fig. 23). Fitting was performed with Matlab (Mathworks), using the 'fit' function. The precision of the fit can be determined by the 95% confidence bounds. See Supplementary Note 9 for more detail.

**Data availability.** The data that support the findings of this study are available from the corresponding author on reasonable request.

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

## Acknowledgements

This work was funded in part by grants from the Claire Giannini Fund, the United States Air Force (FA9550-15-1-0007), the National Institutes of Health (NIH DP50D012179), the National Science Foundation (NSF 1438340), the Damon Runyon Cancer Research Foundation (DFS#06-13), the Susan G. Komen Breast Cancer Foundation (SAB15-00003), the Mary Kay Foundation (017-14), the Donald E. and Delia B. Baxter Foundation, the Skippy Frank Foundation, the Center for Cancer Nanotechnology Excellence and Translation (CCNE-T; NIH-NCI U54CA151459) and the Stanford Bio-X Interdisciplinary Initiative Program (IIP6-43). A.d.l.Z is a Chan Zuckerberg Biohub investigator and a Pew-Stewart Scholar for Cancer Research supported by The Pew Charitable Trusts and The Alexander and Margaret Stewart Trust. O.L. is grateful for a Stanford Bowes Bio-X Graduate Fellowship. E.D.S. wishes to acknowledge funding from the Stanford Biophysics Program training grant (T32 GM-08294). We wish to thank Dr Joseph M. Kahn and Dr Joseph W. Goodman for insightful discussions, Roopa Dalal for images of tissue sections, Ayana Henderson, Nicholas Dwork and Yonatan Winetraub for useful discussions, Timothy R. Brand and the Ginzton Crystal Shop for creating the lapped diffuser, Stanford Neuroscience Microscopy Service (supported by NIH NS069375) and Jim Strommer for custom artwork in Fig. 1. We appreciate the help of Dr Audrey (Ellerbee) Bowden and her laboratory, especially Gennifer Smith, for help with creating phantoms and useful discussions.

## Author contributions

O.L., M.D.L., E.D.S., S.C. and A.d.l.Z. conceived and designed the research; O.L., M.D.L., E.D.S., R.D. and D.S. performed experiments; D.M.M. contributed tools and expert advice; O.L. analysed data; O.L., M.D.L., E.D.S., R.D., D.S., D.M.M. and A.d.l.Z. co-wrote the paper.

## Additional information

**Competing interests:** O.L., M.D.L., E.D.S. and A.d.l.Z. are listed as inventors on a USPTO provisional patent application (62/243466) and an international patent application (PCT/US2016/057656) related to the work presented in this manuscript. The remaining authors declare no competing financial interests.

DOI: 10.1038/ncomms16131    **OPEN**

# Erratum: Speckle-modulating optical coherence tomography in living mice and humans

Orly Liba, Matthew D. Lew, Elliott D. SoRelle, Rebecca Dutta, Debasish Sen, Darius M. Moshfeghi, Steven Chu & Adam de la Zerda

*Nature Communications* 8:15845 doi: 10.1038/ncomms15845 (2017); Published 20 Jun 2017; Updated 11 Jul 2017

An incorrect version of the Supplementary Information was inadvertently published with this Article which included incorrect figure references in Supplementary Table 4. The Article has now been updated to include the correct version of the Supplementary Information.

