## [Peer Review File · Nature Communications]

Editorial Note:

This manuscript has been previously reviewed at another journal that is not operating a transparent peer review scheme. This document only contains reviewer comments and rebuttal letters for versions considered at Nature Communications.

Reviewers' comments:

Reviewer #4 (Remarks to the Author):

The manuscript is improved and addresses many reviewer concerns.

Some remaining concerns I have are:

One limitation of the approach that is not fully discussed is that the sample is illuminated with a speckled field. This is the consequence of passing the sample illumination through a diffuser. Thus, in contrast to angular compounding methods, the use of a rotating diffuser introduces a new source of noise (uncertainty) into the image process. This noise is uncertainty in illumination intensity, which transitions from shot-noise statistics to speckle statistics after passage through the diffuser. This is noted in Figs. 1-2 in "Speckle-field digital holographic microscopy" (Park, et al., Optics Express, 2009). In this context, averaging is required not only to eliminate speckle in the sample but also to eliminate speckle in the sample illumination. To the point of the claim of the presented method being "speckle-free," a 66% reduction in speckle was observed. In the reply to reviewer questions, the authors attribute residual local variation in pixel values not to the method but rather to "the signal variation due to the non-uniformity of the sample." Is it also possible that this residual variation is due to the nature of speckled illumination itself? After all, if 100 independent frames are averaged, that means that the effective speckle contrast of the illumination (even before the illumination is scattered by the sample) is 10%. To that point, it would be instructive to use the proposed SFOCT system to image a flat surface such as a mirror, a surface that would not generate speckle. Would not a single (un-averaged) SFOCT 2D image of a flat mirror be speckled because the sample illumination is cycling through different speckle patterns as the diffuser rotates?

As such, I'm skeptical of the claim that the images presented are fully speckle-free. The authors assert that "the signal variation due to the non-uniformity of the sample...hence, the scattering from different voxels in the sample will not be uniform." While true, this is a minor source of signal fluctuation (e.g. this term is neglected in dynamic light scattering OCT work [Lee, Boas, et al., Optics Express, v20, p22262]). To the extent there may be variation, it is exacerbated by the fact that gold nanorods are used, which have anisotropic scattering properties, so that there is an additional degree of freedom between different voxels in the phantom (rotational position of the gold nanorods).

In terms of the prior angular compounding literature (e.g. Desjardins, et al.), one thing to note about these papers is that signal was not collected over all available solid angles but rather solid angles that intersected with a single effective detection plane (i.e. a planar angle). That is, speckle patterns were collected only within one planar angle allowed by a lens, not over the entire solid angle allowed by a lens. Thus, at least in principle, the performance of angular compounding is higher than reported in these manuscripts because there is a rotational degree of freedom present that these authors did not exploit.

The authors comment that: "As noted in the discussion of the manuscript, applying SFOCT for imaging

fast moving objects may be feasible by averaging A-scans instead of frames (as we demonstrated in this manuscript) or by using novel compounding methods such as interleaved OCT." However, not discussed is whether or not any diffuser can rotate through ~ 10 to 100 (or more) independent speckle patterns over the course of a single A-scan. Even if we assume lower limits (10 kHz effective A-scan rate, 10 independent speckle cells per effective A-scan), the diffuser would have to cycle through 100,000 speckle cells per second. Is this feasible?

Speckle-Modulating Optical Coherence Tomography in Living Mice and Humans:

Point-by-point response

We thank the reviewer and the editor for their valuable comments and suggestions. We have made additional improvements to the manuscript and added several results following the reviewer's latest questions. As a note, the method we describe for speckle removal has been an indispensable technique for several ongoing *in vivo* studies in our lab. We expect the same will be true for many researchers using OCT.

In response to the reviewer's current comments, we have changed the name of the method from Speckle-Free OCT (SFOCT) to **Speckle-Modulating OCT (SMOCT)**. This new name describes the speckle-modulating system, rather than the speckle-free outcome, since a complete removal of speckle would require acquisition of an unlimited number of frames and is not practical.

Author responses are in blue, changes and additions to the manuscript are in magenta, all line numbers refer to the documents with highlights and "Track changes".

Reviewer #4 (Remarks to the Author):

The manuscript is improved and addresses many reviewer concerns.

Thank you for appreciating the improvements of the manuscript. We are grateful for the insightful comments and suggestions of the reviewer that helped improve it.

Some remaining concerns I have are:

One limitation of the approach that is not fully discussed is that the sample is illuminated with a speckled field. This is the consequence of passing the sample illumination through a diffuser. Thus, in contrast to angular compounding methods, the use of a rotating diffuser introduces a new source of noise (uncertainty) into the image process. This noise is uncertainty in illumination intensity, which transitions from shot-noise statistics to speckle statistics after passage through the diffuser. This is noted in Figs. 1-2 in "Speckle-field digital holographic microscopy" (Park, et al., Optics Express, 2009). In this context, averaging is required not only to eliminate speckle in the sample but also to eliminate speckle in the sample illumination.

To the point of the claim of the presented method being "speckle-free," a 66% reduction in speckle was observed. In the reply to reviewer questions, the authors attribute residual local variation in pixel values not to the method but rather to "the signal variation due to the non-uniformity of the sample." Is it also possible that this residual variation is due to the nature of speckled illumination itself?

After all, if 100 independent frames are averaged, that means that the effective speckle contrast of the illumination (even before the illumination is scattered by the sample) is 10%. To that point, it would be instructive to use the proposed SFOCT system to image a flat surface such as a mirror, a surface that would not generate speckle. Would not a single (un-averaged) SFOCT 2D image of a flat mirror be speckled because the sample illumination is cycling through different speckle patterns as the diffuser rotates?

We thank the reviewer for pointing out the introduction of speckle illumination by the diffuser. Indeed, the diffuser creates variability in the illumination intensity, as can be seen in Supplementary Fig. 21 (previously Supplementary Fig. 20). Following the reviewer's suggestions, we have now added a new Supplementary Note (Supplementary Note 5) with a characterization of the variability in the illumination and its reduction by averaging. The variability in illumination intensity can be reduced by averaging over scans that are acquired with different diffuser positions, which is also needed to remove speckle noise. The removal of speckle illumination by averaging is also reported in the paper mentioned by the reviewer (Park, et al., Optics Express, 2009¹): "We note that there have been studies in which time varying speckles were used for HPM to eliminate non-uniform intensity distribution²⁻⁴". Park et al also mention the reduction of diffraction artifacts by speckle-illumination ("As a result, the transmitted light becomes incoherent in time and space, which lead to improvement in image performance: reducing the effect of unwanted diffractions..."), which is also visible in the results we obtained. As shown in the new Supplementary Fig. 4 (copied below), the variability in the normalized intensity is reduced in SMOCT (formerly SFOCT) compared to OCT owing to the reduction of diffraction artifacts, which are visible in the OCT image as circular Airy patterns that are likely due to dust particles.

Supplementary Figure 4 | Characterization of the illumination variability originating from the diffuser. *en face* images of a glass slide (averaged over the depth and 100 frames, shown in a similar logarithmic scale) scanned with (a) no diffuser, (b) a static diffuser and (c) a moving diffuser. Scale bar is 100 μm . The image with the static diffuser (b) shows the variability in the illumination of the sample due to the diffuser. The SMOCT image (c), which was acquired with a moving diffuser, shows the reduction in illumination variability obtained with averaging. d, Normalized histograms of the normalized pixel intensity (the signal of each pixel divided by

average signal in the image, in linear scale) for the images shown in (a-c). The histograms show that the illumination statistics changes from a Rayleigh distribution to a much narrower distribution following averaging over 100 frames, indicating that on average the illumination-uniformity of the sample is improved. Furthermore, the diffuser was able to remove diffraction artifacts that broadened the intensity distribution in the OCT image, therefore, overall, SMOCT was able to improve the illumination-uniformity compared to OCT. The standard deviation of the normalized pixel intensities is 0.12, 0.53 and 0.06 for OCT, static diffuser, and SMOCT, respectively.

It is interesting to point out that by introducing varying external speckle noise (“speckle illumination”), we are able to remove the internal speckle noise, which originates from within the sample.

With regards to the question of the reviewer: “Is it also possible that this residual variation is due to the nature of speckled illumination itself?”, the answer is yes - there is a residual variation due to the speckle illumination. We have now added this observation to the Main Text (lines 165-168):

“Additional sources for signal variability in the SMOCT image are the absorption of the sample, size variability of the scattering nanoparticles, distance from the focal plane and residual illumination variability, which was created by the diffuser and is characterized in Supplementary Note 5 and Supplementary Fig. 4-5.”

Additionally, we are able to reduce the artifacts of residual speckle illumination in post-processing. The variability in illumination manifests as darker and brighter A-scans, which are consistent through the depth of the sample. If there is remaining variability, which may occur when averaging is limited, we can compensate for it by amplifying each A-scan adaptively, by assuming that the average OCT signal of the sample does not change very quickly. We have now added this post-processing algorithm and its demonstration to Supplementary Note 5 and Supplementary Fig. 5 (also copied here). Please note that this post-processing method was not applied to any of the images in the manuscript. We found that algorithmic removal of the signal variation due to speckle illumination was only needed when performing fewer than 20 A-scan averages.

“If there is remaining variability, we can compensate for it by amplifying each A-scan adaptively, by assuming that the average signal across the sample does not change very quickly. The algorithm used for postprocessing is described below:

$$\bar{I}(x_n) = \frac{1}{M} \sum_{m=1}^M I^{initial}(x_n, z_m) \quad (\text{Eq. S6a})$$

$$I^{final}(x_n, z_m) = I^{initial}(x_n, z_m) \times \frac{LPF\{\bar{I}(x_n)\}}{\bar{I}(x_n)} \quad (\text{Eq. S6b})$$

x_n and z_m denote the discretized locations in the images along the lateral and axial dimensions, respectively. $I^{initial}(x_n, z_m)$ is the initial signal, in linear scale, after averaging A-scans and/or frames, $\bar{I}(x_n)$ is the average signal along the depth of the

image or a region inside the sample and $I^{final}(x_n, z_m)$ is the post-processed image. The LPF operator represents a low-pass filter, that can be implemented as a convolution with a Gaussian kernel or a median filter. This algorithm can be easily extended to a 3D volume by low-pass filtering the intensity of the 2D projection and by applying the adaptive gain to each A-scan in the volume.

In Supplementary Fig. 5 we compare SMOCT scans of a PDMS + TiO₂ phantom with 20 A-scan averages, before and after compensation, to OCT and SMOCT scans with 100 frame averages. The residual vertical lines (Supplementary Fig. 5d) are not visible after postprocessing (Supplementary Fig. 5e) and the TiO₂ aggregates can be observed. Low pass filtering was implemented by a median filter with a 50 pixel window.”

Supplementary Figure 5 | Residual illumination variability and its reduction in post-processing. **a**, OCT B-scan of a phantom composed of PDMS and TiO₂ powder. **b**, A close-up view on the region shown in (a). **c**, The same region, acquired with a static diffuser. **d**, SMOCT image with 20 A-scan averages. Speckle noise is reduced; however, the image shows vertical line artifacts due to illumination variability that is not entirely removed by averaging. **e**, The image in (d), after correction of the vertical-line artifact in post-processing. **f**, SMOCT image with 100 B-scan (frame) averages and a moving diffuser. The vertical line artifacts are not visible and speckle noise is significantly reduced, revealing the variable distribution of TiO₂ in the phantom.

As such, I'm skeptical of the claim that the images presented are fully speckle-free.

We appreciate the reviewer's comment and acknowledge that technically the images in our manuscript are not fully speckle-free. Our method enables acquisition of an unlimited number of frames (or A-scans) with un-correlated speckle which, when averaged, **can remove speckle noise arbitrarily well**. Indeed, fully removing speckle noise (originating from the sample and from the illumination through diffuser) would require an infinite number of frames, which is not practical. That said, our method is able to remove significantly more speckle noise from OCT volumes compared to previously reported methods, and as demonstrated in the manuscript's figures, SMOCT reveals structures in living tissue, thus improving the performance of OCT in practical applications.

Owing to the practical limitation of achieving fully speckle-free images, we have now changed the name of our method from Speckle-Free OCT (SFOCT) to **Speckle-Modulating OCT (SMOCT)**. This new name describes the speckle-modulating system, rather than the speckle-free outcome, since a complete removal of speckle would require acquisition of an unlimited number of frames and is not technically feasible. We have updated the title and acronym throughout the manuscript accordingly and revised the wording to convey that speckle noise can be removed arbitrarily well and not entirely (lines 19, 236, 260, 261, 275, 284-285 of the Main Text).

The authors assert that "the signal variation due to the non-uniformity of the sample...hence, the scattering from different voxels in the sample will not be uniform." While true, this is a minor source of signal fluctuation (e.g. this term is neglected in dynamic light scattering OCT work [Lee, Boas, et al., Optics Express, v20, p22262]). To the extent there may be variation, it is exacerbated by the fact that gold nanorods are used, which have anisotropic scattering properties, so that there is an additional degree of freedom between different voxels in the phantom (rotational position of the gold nanorods).

We thank the reviewer for the insightful comment regarding the additional degree of freedom causing signal variability in the gold-nanorod phantom. We have now created and measured a new phantom, based on isotropic gold nanospheres (GNSs) in agarose. With this phantom, the inherent variability was indeed reduced and the speckle contrast we measured with SMOCT was reduced by 77.8% (compared to 66% with the gold nanorod phantom). Following these new measurements, we have now replaced Fig. 2 of our manuscript entirely (copied below) and also updated the relevant text in the Main, Methods and Supplementary documents. As noted previously in this response, we attribute the residual contrast to the random distribution of nanoparticles in sample, the absorption of the sample, size variability of the scattering nanoparticles, distance from the focal plane and residual illumination variability.

Figure 1 | Demonstration of speckle removal by analysis of the speckle statistics and speckle contrast. **a, b**, OCT and SMOCT images of **GNSs** dispersed in agarose (scale bar is 100 μm). The OCT image shows a combination of speckle noise and the signal variation from the random distribution of **GNSs** in the phantom. The SMOCT image shows only the latter. This claim is supported by the statistical analysis of pixel intensities. **c, d**, Statistical analysis of the pixel values shows that the OCT image (**c**) is dominated by speckle noise and the distribution of pixel values is approximately a Rayleigh distribution that persists with averaging (M is number of averages). In SMOCT (**d**), increasing the number of averages narrows the distribution significantly. **e**, Reduction in normalized standard deviation (SD) versus the number of averages, M , for OCT and SMOCT. The reduction in the normalized SD is significantly larger in SMOCT versus OCT. **f**, The reduction of normalized speckle as defined by Supplementary Equation 7 (see Supplementary Note 6) follows $1/\sqrt{M}$, as expected.

It is interesting that Lee, Boas, et al., Optics Express, v20, p22262⁵ neglect the variation in signal due to the random distribution of nanoparticles in the sample. Perhaps this is due to the dynamic nature of the samples they are studying and that diffusion and flow effects are more dominant than the Poisson distribution we observe in the static samples in our manuscript. Because the Poisson distribution is only one of the likely sources of residual signal variability and to avoid the confusion of readers (which was mentioned in the reviewer comments in the previous version), we have now reduced the analysis of the Poisson distribution in our manuscript.

In Fig. 3 of the Main Text, we present a phantom composed of polystyrene beads and large gold nanorods (LGNRs). Following the reviewer's comment, we have now added this sentence to the Main Text (lines 188-190):

“Conversely, SMOCT enabled detection of the beads along with the random signal originating from the LGNRs, which is influenced by their random distribution and orientation (Fig. 3c,e,g).”

In terms of the prior angular compounding literature (e.g. Desjardins, et al.), one thing to note about these papers is that signal was not collected over all available solid angles but rather solid angles that intersected with a single effective detection plane (i.e. a planar angle). That is, speckle patterns were collected only within one planar angle allowed by a lens, not over the entire solid angle allowed by a lens. Thus, at least in principle, the performance of angular compounding is higher than reported in these manuscripts because there is a rotational degree of freedom present that these authors did not exploit.

We thank the reviewer for the note on angular compounding. Indeed, angular compounding has the potential to substantially remove speckle, and much more if applied on the solid angle compared to one plane. However, the application of angular compounding over a solid angle is still limited in the number of uncorrelated speckle patterns it can produce (due to the finite number of non-overlapping beams that can fit inside the pupil of the lens) and, to our knowledge, has not been demonstrated. In addition, the implementation of planar angular compounding (as demonstrated, for example, in the publications by Desjardins et al.^{6,7}) requires synchronized mirrors in the illumination and the collection light-paths. Implementation of such angular compounding over the solid angle may be very technically challenging.

We would like to mention that SMOCT offers an unlimited number of uncorrelated speckle patterns owing to the random illumination through the diffuser, enabling an arbitrarily large reduction of speckle noise. Furthermore, SMOCT can be applied as a rather simple add-on to existing systems, at a price of about \$200-\$300 for the diffuser and rotator (the lenses for the 4f imaging system are an additional cost, however, they may not be needed in all systems, such as those used in ophthalmic devices).

Following the reviewer’s comment on performing angular compounding on the solid angle, we have now added the following sentences (shown in magenta) to Supplementary Note 12, which compares several methods of speckle reduction:

“Angular compounding often produces favorable results, however it is limited in its capability to reduce speckle due to the limited number of obtainable uncorrelated speckle patterns as a result of the overlap of the beams illuminating the sample at different angles⁶⁻⁹. Desjardins et al⁶ have demonstrated speckle noise removal that is equivalent to compounding 30 uncorrelated speckle patterns, which provides a substantial improvement in image quality and can theoretically be extended if applied on the solid angle instead of a single plane. However, such removal is still limited in the number of uncorrelated speckle patterns it can produce.”

The authors comment that: "As noted in the discussion of the manuscript, applying SFOCT for imaging fast moving objects may be feasible by averaging A-scans instead of frames (as we demonstrated in this manuscript) or by using novel compounding methods such as interleaved OCT." However, not discussed is whether or not any diffuser can rotate through ~10 to 100 (or more) independent speckle patterns over the course of a single A-scan. Even if we assume lower limits (10 kHz effective A-scan rate, 10 independent speckle cells per effective A-scan), the diffuser would have to cycle through 100,000 speckle cells per second. Is this feasible?

We thank the reviewer for this question. We don't expect that the rotation speed of the diffuser will pose a practical limit to acquiring multiple independent speckle patterns within an effective A-scan (where "effective A-scan" refers to several A-scans acquired at the same location and averaged into a single A-scan in the resulting image). Our current rotational stage (RSC-103, Pacific Laser Equipment) has a tangential velocity of 9 mm/s at the edge of the diffuser. We have shown in Supplementary Figure 14 that using the 1500 grit diffuser the speckle patterns are decorrelated within one A-scan. At an A-scan rate of about 20 kHz (meaning, 20,000 speckle cells per second), this decorrelation corresponds to a "speckle cell" of 0.45 μm at most. There are several ways to increase the rate of "speckle cells" per second:

- It is possible that a "speckle cell" is smaller than what we were able to measure (due to the limited A-scan rate of our OCT system). In this case, it would be possible to obtain increased speckle reduction at an increased A-scan rate, without changing the diffuser or the rotational stage.
- The tangential speed of the diffuser can be increased by enlarging the diffuser. Since the tangential velocity is proportional to the radius, using a 2" diffuser and rotator (which is also available from Pacific Laser Equipment) would result in doubling the rate of "speckle cells" per second. Using larger diffusers may also be possible.
- Using a different controller or a different motor would result in faster rotation. There are faster motors (from Thorlabs, Newport, Aerotech, and maybe others), however, they cost more than the motor used in this study.
- One could use methods such as interleaved OCT¹⁰ to acquire several A-scans in parallel, which would increase the effective A-scan rate while allowing acquisition of additional uncorrelated speckle patterns.
- The roughness pattern of the diffuser influences the size of the "speckle cell". It may be feasible to create a diffuser with a small lateral feature size but with enough thickness variation in order to minimize the movement needed by the diffuser to achieve speckle decorrelation.

These could be combined to achieve a very fast change of "speckle cells". We have now added these options for increasing the rate of speckle decorrelation to the Discussion of the Main Text (lines 300-304, only the first sentence is new):

“If an OCT with a very fast A-scan rate is used, the rotation speed of the diffuser can be increased by using a faster motor or a larger diffuser, since the tangential velocity is proportional to the diffuser radius. Another way to acquire SMOCT images of fast-moving objects is by implementing a conventional tissue-tracking system¹¹ or a system that can achieve image compounding without extending the acquisition time, such as interleaved OCT¹⁰.”

Please note that that the speed of the diffuser should not be much faster than the acquisition speed of the OCT in order to prevent washout of the interference fringes, as noted in the Main Text (lines 238 and 299).

Bibliography to this response:

1. Park, Y. *et al.* Speckle-field digital holographic microscopy. *Opt. Express* **17**, 12285 (2009).
2. Somekh, M. ., See, C. . & Goh, J. Wide field amplitude and phase confocal microscope with speckle illumination. *Opt. Commun.* **174**, 75–80 (2000).
3. Pitter, M. C., See, C. W. & Somekh, M. G. Full-field heterodyne interference microscope with spatially incoherent illumination. *Opt. Lett.* **29**, 1200 (2004).
4. Dubois, F., Novella Requena, M.-L., Minetti, C., Monnom, O. & Istasse, E. Partial spatial coherence effects in digital holographic microscopy with a laser source. *Appl. Opt.* **43**, 1131 (2004).
5. Lee, J., Wu, W., Jiang, J. Y., Zhu, B. & Boas, D. A. Dynamic light scattering optical coherence tomography. *Opt. Express* **20**, 22262 (2012).
6. Desjardins, A. E., Vakoc, B. J., Tearney, G. J. & Bouma, B. E. Speckle Reduction in OCT using Massively-Parallel Detection and Frequency-Domain Ranging. *Opt. Express* **14**, 4736 (2006).
7. Desjardins, A. E. *et al.* Angle-resolved Optical Coherence Tomography with sequential angular selectivity for speckle reduction. *Opt. Express* **15**, 6200 (2007).
8. Iftimia, N., Bouma, B. E. & Tearney, G. J. Speckle reduction in optical coherence tomography by ‘path length encoded’ angular compounding. *J. Biomed. Opt.* **8**, 260–3 (2003).
9. Hughes, M., Spring, M. & Podoleanu, A. Speckle noise reduction in optical coherence tomography of paint layers. *Appl. Opt.* **49**, 99 (2010).
10. Duan, L. *et al.* Single-shot speckle noise reduction by interleaved optical coherence tomography. *J. Biomed. Opt.* **19**, 120501 (2014).
11. Braaf, B. *et al.* Real-time eye motion correction in phase-resolved OCT angiography with tracking SLO. *Biomed. Opt. Express* **4**, 51–65 (2013).

REVIEWERS' COMMENTS:

Reviewer #4 (Remarks to the Author):

The revised manuscript is improved and was responsive to reviewer feedback. I have no further feedback.

Speckle-Modulating Optical Coherence Tomography in Living Mice and Humans:

Point-by-point response

Reviewer #4 (Remarks to the Author): (Author responses are in blue)

The revised manuscript is improved and was responsive to reviewer feedback. I have no further feedback.

We thank the reviewer for the insightful comments that have helped improve the manuscript in the previous revisions. We have not made any changes to the content of the manuscript in this revision.